# In Silico Comparative Exploration of Allergens of *Periplaneta americana*, *Blattella germanica* and *Phoenix dactylifera* for the Diagnosis of Patients Suffering from IgE-Mediated Allergic Respiratory Diseases

**DOI:** 10.3390/molecules27248740

**Published:** 2022-12-09

**Authors:** Mohd Adnan Kausar, Tulika Bhardwaj, Sadaf Anwar, Fahaad Alenazi, Abrar Ali, Khalid Farhan Alshammari, Shimaa Mohammed Hasnin AboElnaga, Rajeev Singh, Mohammad Zeeshan Najm

**Affiliations:** 1Department of Biochemistry, College of Medicine, University of Ha’il, Ha’il 2440, Saudi Arabia; 2Department Agricultural Food and Nutritional Sciences, University of Alberta, Edmonton, AB T6G 2P5, Canada; 3Department of Pharmacology, College of Medicine, University of Ha’il, Ha’il 2440, Saudi Arabia; 4Department of Ophthalmology, College of Medicine, University of Ha’il, Ha’il 2440, Saudi Arabia; 5Department of Internal Medicine, College of Medicine at University of Ha’il, Ha’il 2440, Saudi Arabia; 6Department of Basic Science, Deanship of Preparatory Year, University of Hail, Ha’il 2440, Saudi Arabia; 7Department of Environmental Studies, Satyawati College, University of Delhi, Ashok Vihar III, Delhi 110052, India; 8Department of Environmental Science, Jamia Millia Islamia (Central University), New Delhi 110025, India; 9School of Biosciences, Apeejay Stya University, Sohna, Gurugram 122003, India

**Keywords:** allergens, IgE, respiratory allergy, *Periplaneta americana*, *Blattella germanica*

## Abstract

The burden of allergic illnesses is continuously rising, and patient diagnosis is a significant problem because of how intricately hereditary and environmental variables interact. The past three to four decades have seen an outbreak of allergies in high-income countries. According to reports on the illness, asthma affects around 300 million individuals worldwide. Identifying clinically important allergens for the accurate classification of IgE-mediated allergy respiratory disease diagnosis would be beneficial for implementing standardized allergen-associated therapy. Therefore, the current study includes an in silico analysis to identify potential IgE-mediated allergens in date palms and cockroaches. Such an immunoinformatic approach aids the prioritization of allergens with probable involvement in IgE-mediated allergic respiratory diseases. Immunoglobulin E (IgE) was used for molecular dynamic simulations, antigen–antibody docking analyses, epitope identifications, and characterizations. The potential of these allergens (Per a7, Per a 1.0102, and Bla g 1.0101) in IgE-mediated allergic respiratory diseases was explored through the evaluation of physicochemical characteristics, interaction observations, docking, and molecular dynamics simulations for drug and vaccine development.

## 1. Introduction

Asthma and allergic rhinitis are two of the most prevalent respiratory allergies worldwide, and their prevalence is steadily rising. Traditional ways of life and the environments in which people live have an impact on the prevalence of asthma [1,2,3]. According to respiratory allergy data from the Kingdom of Saudi Arabia, allergic rhinitis and bronchial asthma are present in 13.5% and 11.2% of the population, respectively [4]. German cockroaches are some of the most common indoor allergens in countries that traditionally feed on floors, such as Saudi Arabia and other Gulf countries [5,6,7,8]. As a result of improved illness characterization, particularly through the application of cutting-edge technologies, including “omics,” it has become clear that many subgroups exist within various disease entities, such as asthma, allergic rhinitis, atopic dermatitis (AD), and angioedema (AE) [9,10]. In the current work, we attempted to explore several significant cockroach (CR) allergenic elements. Allergens are non-parasite proteins or molecules linked to proteins that cause atopic individuals to produce large amounts of IgE [11]. In nonatopic patients, these substances cause the development of alternative immunoglobulin isotypes, such as IgM and/or IgG, with little to no IgE [12,13].

CR allergens are glycosylated antigenic proteins with molecular weights of 6 to 120 kDa that exist in multimeric forms [14] and disintegrate in the human body. CR allergens can be found in several stages of insect life cycles and development, including in the cuticle, dead body debris, eggs, and egg casts [15]. Hemolymph, regurgitating fluid, urine, faeces, and body washes are all fluids that can contain parasites [16]. CR allergenic particles (>10 microns) are frequently present on surfaces such as floors, lamps, and tables, but the wind quickly disturbs and disperses them. American cockroach (*Periplaneta americana*) and German cockroach (*Blattella germanica*) allergens are the most common in Saudi households.

Bla g 1, Bla g 2, Bla g 4, Bla g 5, Bla g 6, and Bla g 7 (Per g = genus species; numerical value = the number of allergens from that species) are allergens from *Blattella germanica*. The allergens from *Periplaneta americana* are Per 1, Per 2, Per 3, Per 4, Per 6, Per 7, Per 9, and Per 10 [17]. With the help of a benzamidine sepharose column and immunological and biochemical analysis, Per a 10 was isolated from American CR extract, a serine proteinase, the molecular weight of which is thought to be around 28 kDa. Skin tests and immunoblots showed IgE reactivity in >80% of cockroach-sensitized patients, identifying Per a 10 as a significant allergen. According to the latest study, Per a 10 can increase the phenotype of dendritic cell type 2 by upregulating CD86, increasing high interleukin-6 (IL-6) secretion, and decreasing IL-12 secretion, in addition to changing the expression of CD40 on the dendritic cell surface via the nuclear factor-B (NF-B) pathway [18,19].

The *in-silico* method was used in this study to identify potential IgE-mediated allergens in date palms and cockroaches [20]. Therefore, epitope identification and characterization were performed to give future researchers a platform to develop potential vaccines against allergies caused by the allergens under investigation. In addition, antigen–antibody docking and molecular dynamic simulations with IgE were performed to assess the allergenic potential for future research.

## 2. Methodology

### 2.1. Retrieval of the Protein Sequences of Date Palm and Cockroach Allergens

The sequences of allergen proteins were obtained from the UniProt online database (https://www.uniprot.org/help/uniprotkb, accessed on 24 May 2022), which is the major archive for different types of repositories used to carry out comprehensive genomic and proteomic analyses.

### 2.2. Physiochemical Parameter Evaluations of the above Allergens

The proteins identified as possessing physical and chemical characteristics were investigated for physiochemical properties, and the ProtParam server [21] was used to determine theoretical parameters, such as molecular weights, amino acids, pI (isoelectric point) values, instability indices, etc.

### 2.3. Functional Classifications

The primary identification mechanism for understanding pathogenesis in organisms is the distinction between virulent and non-virulent proteins. The VICMpred online prediction server [22] was used for the functional classification of bacteria using a bi-layer cascade SVM approach, which applies sequence information for the prediction of different virulence factors. The VICMpred webserver uses amino acid sequences in pattern-based approaches that show extremely important values of functional classification, i.e., median values >1.0.

### 2.4. Subcellular Localization

Protein localization is a significant aspect in the development of new drug targets for drug sightings, as cytoplasmic and membrane proteins have been recognized as pharmacological targets. Since no information regarding the subcellular localization of these protein sequences was available at the time, Plant-mSubP, a two-level support vector machine tool [23] for the prediction of subcellular localizations of single and multiple protein sequences, was utilized for *Phoenix dactylifera.* In the case of American cockroach (*Periplaneta americana*) and German cockroach (*Blattella germanica*) allergens, WoLF-PSORT [24] enabled subcellular prediction based on sorting signals, amino acid compositions, and functional motifs, such as DNA-binding motifs. More than one software package was utilized for accurate computational identifications of subcellular localizations.

### 2.5. Prediction of IgE Epitopes and Allergenic Site Prediction

The AlgPred server [25] was used to ensure the allergenicity potential of the protein sequences. It includes the integrated method of combining SVM amino acid composition or dipeptide-based methods, IgE epitope mapping, BLAST searching against allergen representative peptides (ARPs), and MAST (Motif Alignment and Search Tool)–MEME (Multiple EM for Motif Elicitation) suites to measure putative allergenicity for default parameters [26]. Further, AllerCatPro 2.0 [27] predicts the allergenic potential of protein sequences based on the structural similarities of their three-dimensional structures and their amino acid compositions when compared with protein allergens derived from public repositories. IgE sensitization towards proteins is frequently recognized upon exposure to aeroallergens, food allergens, and personal care products [28].

### 2.6. Secondary Structure Prediction

The SOPMA web server [29] was used to predict the 2D structures of target protein sequences. This online server enables simple and accurate predictions for the identification of different forms of a characteristic in secondary structures, such as alpha helices, beta turns, extended strands, and random coil regions, which contain the primary elements of 2D structure prediction.

### 2.7. Tertiary Structure Prediction (3D Model) and Validation

Homology modeling was performed using MODELLER to generate the 3D molecular structures of identified stable protein sequences of date palm and cockroach allergens that contain experimentally proven IgE epitopes [30,31]. MODELLER is a computational platform for comparative protein structure modeling which can be used to generate tertiary protein models [32].

### 2.8. Antigen–Antibody Docking Studies

Molecular docking analysis of all the prioritized antigenic sites and IgE was performed to estimate binding affinities. The human IgE three-dimensional structure was retrieved from RCSPDB (PDB ID: 4J4P and UniProt ID: P01854). The structure includes a complex of Human IgE-Fc with two bound Fab fragments. UCSF Chimera (https://www.cgl.ucsf.edu/chimera/, accessed on on 24 May 2022) enables manual preprocessing of these peptides, further homodimers were reduced to a single chain to reduce docking time, and non-amino acid molecules (ligands, ions, and solvent water) were removed to prevent hindrances during docking. Rigid-body molecular docking of the engineered vaccine with the processed receptors was performed based on shape-complementarity principles, utilizing PatchDock server [33]. This server differentiates a protein’s surface into small patches (convex, concave, and flat) using a segmentation algorithm that is superposed using a shape-matching algorithm. The top conformations obtained with PatchDock were subjected to docking score refinement using FireDock [34]. FireDock refined the docked poses by optimizing side-chain conformations and rigid body orientations via Monte Carlo simulation.

### 2.9. MD Simulation

To analyze conformational stability, molecular dynamics and simulation studies were performed using GROMACS v5.1.5 and the OPLS-AA/L all-atom force field (2001 amino acid dihedrals). To study the interfacial atoms’ physical movements and the complexes’ stabilities in explicit water boxes (dodecahedrons), docked complexes were subjected to 309 K for 25 ns [35,36]. We used both NVT and NPT ensembles to mimic real experimental conditions. A six-step procedure was followed to analyze the trajectories of energy minimization in the MD simulations: (a) energy minimization of solvent molecules prior to the entire system using the Broyden–Fletcher–Goldfarb–Shanno (LBFGS) algorithm; (b) non-hydrogen solute atoms were restricted to 300 K temperature and 1 bar for 40 ns to attain equilibrium states; (c) control of temperature and pressures during initial simulations using Berendsen thermostats and the barostat algorithm; (d) initial trajectories were obtained to assist in the analysis of RMSDs; (e) RMSF hydrogen bond analysis was performed; and (f) determination of the radii of gyration for the antigen–antibody docked complexes [37,38].

## 3. Results and Discussion

### 3.1. Retrieval of the Protein Sequences for Date Palm and Cockroach Allergens

UniProt, a freely available database of protein sequences and their functional annotations from several genome sequencing projects, was utilized to mine the primary dataset protein sequences of the date palm and cockroach allergens. The primary dataset was then subjected to physiochemical characterization to evaluate pathogenic and allergenicity potential.

### 3.2. Physiochemical Parameter Evaluation of the above Allergens

ProtParam enabled the theoretical computation of instability indices, molecular weights, and GRAVYs of the identified allergens of date palms and cockroaches (Table 1). An instability index estimates the stability of the protein in a test tube. The sum of hydropathy values for all amino acids, when divided by the number of residues in a sequence, predicts the GRAVY (grand average of hydropathicity). The relative volume occupied by aliphatic side chains (alanine, valine, isoleucine, and leucine) in a protein sequence is termed an aliphatic index. The amino acid residue contents of the screened allergens ranged between 151 and 823. The instability index can be used to infer the stability of a measured protein in a test tube. Therefore, in this study, only stable amino acids with instability indices <40 were selected for further allergenicity and functional predictions.

### 3.3. Functional Characterization

VICMpred enables the functional characterization of iterated protein sequences into major functional modules (Table 2). This machine-learning-based tool identified the active participation of primary protein sequences in metabolism molecules (32.35%), virulence factors (5.88%), cellular processes (44.12%), and information and storage (17.65%). Functional characterization is important in proteomics studies to understand biological functions at the system level.

### 3.4. Subcellular Localization

Plant-mSubP and WoLF-PSORT both assist in the selective distribution of primary protein sequences within cellular compartments. The sorting of protein sequences was mainly in the cell membranes, extracellular matrices, mitochondria, and cytoplasmic domains (Appendix A). The prioritized subcellular localizations after analysis were in the cytoplasm, mitochondria, plastids, and vacuoles.

### 3.5. Prediction of IgE Epitopes and Allergenic Site Prediction

Prediction of allergenic proteins and mapping of IgE epitopes by AlgPred identified allergenic peptides by utilizing five approaches (IgE epitope + ARPs BLAST + MAST + SVM) to attain an overall accuracy of 85%. The protein sequences were predicted to be non-allergenic if the score was estimated in the -ve sign (Table 3). The results highlighted that the majority of inputted protein sequences did not contain an experimentally proven IgE epitope. A total of nine protein sequences were screened for allergenicity potential. Further, AllerCatPro 2.0 also validated the allergenicity potential of inputted protein sequences by estimating strong evidence (Table 4), which is a result of higher similarity search values against allergens present in public repositories.

### 3.6. Secondary Structure Prediction

In the absence of a three-dimensional structure and template sequence, secondary structure analysis is necessary to identify the percentages of alpha helices, extended strands, beta turns, and random coils in protein sequences (Table 5). Such analysis helps in the generation of the 3D structures of proteins. This analysis also renders information about protein activities, relationships, and functions. This study identified that alpha helices accounted for the majority coverage of the screened allergens, followed by random coils, beta turns, and extended strands. This implies that hydrogens bonds are mainly responsible for protein–protein interactions in further molecular docking procedures.

### 3.7. Tertiary Structure Prediction (3D model) and Validation

Three-dimensional structures of protein sequences (Bla g 1.0101, accession number: AF072219.2; Per a 7, accession number: ACS14052.1; and Per a 7.0102, accession number: AF106961.1) were mined from the UniProt database. MODELLER was then utilized to generate the three-dimensional structures of the protein sequences. These aided in the identification of template sequences with PDB codes 4JRB, 7KO4, and 6X5Z. Further, validation was performed using the SAVES server. We used a high-resolution structure refinement method, i.e., ModRefine (https://zhanglab.ccmb.med.umich.edu/ModRefiner/, accessed on 15 June 2022), which improves poor rotamers by simulating both protein backbones and side chains. The tertiary models generated were subjected to molecular docking analysis.

### 3.8. Antigen–Antibody Docking Studies

Protein–protein docking analysis was performed after protein backbone stabilization to determine the binding affinities of the resulting plausible antigenic protein sequences. Visualization of ions, ligands, and other non-amino acid molecules is possible by utilizing the Chimera Visualization tool (Figure 1). Rigid docking of antigenic sequences and antibodies was then carried out using PatchDock, and the poses so generated were sorted with respect to the binding energy functions. The top 10 docking outcomes were pulled and furthered for pose refinement using FireDock. The binding energies of the complexes and those with the lowest binding energies were screened (Blag 1.0101-IgE with −19.08 kcal/mol, Bla g 7-IgE with 11.2 kcal/mol, Per a 1.0101-IgE with −9.22 kcal/mol, Per a 1.0103-IgE with −7.12 kcal/mol, Per a 1.0104-IgE with −8.22 kcal/mol, Per a 1.0201-IgE with −7.76 kcal/mol, Per a 1.0102-IgE with −21.33 kcal/mol, and Per a 7-IgE with −19.71 kcal/mol). The visualization of protein–protein docking validated the major role of hydrogen bonding among protein sequences. The docked complexes with optimal minimum binding energies were considered for the MD simulation platform. In the case of Per a 1.0102-IgE, hydrogen bonds were formed among K90, Y82, N103, and Y105 of Per a 1.0102 with antigen recognition sites of IgE. A155, K160, and Q147 of Per a 7-IgE formed hydrogen bonds with variable regions of IgE. For Bla g 1.0101-IgE, the major interacting partners were Q229, L220, and K223 (Figure 1). Such prioritization and inclusion of a molecular docking approach enabled the identification of potential allergens to be considered for vaccine and drug discovery. For the onset and persistence of most immediate-type allergies and several asthma phenotypes, immunoglobulin E (IgE) is essential. As a result, IgE is a key target for both diagnostic and therapeutic objectives [39]. There are two categories of IgE-binding epitopes: linear (sequential) and conformational (discontinuous). While conformational epitopes are generated by spatially nearby AAs that are far apart in the protein’s AA primary sequence, linear epitopes are continuous AA sequences [40]. Bla g 1.0101 is secreted in the cockroach digestive tract, and sensitization occurs through inhalation of allergen-carrying faecal particles that are released into the environment [41]. Tropomyosins, for example, Per a 7, play a role in muscle contraction [42]. Tropomyosin is a pan-allergen found in the muscles of many animals [43,44]. Initially identified as a major shrimp allergen, it has since been found in a variety of insects and causes IgE cross-reactivity [45]. A study using RNA interference-mediated knockdown of this allergen in *Periplaneta americana* confirmed that Per a 1 is involved in digestion and nutrient absorption [43].

### 3.9. MD Simulation

MD simulations of the selected complexes for 25 ns using GROMACS v5.1.2 were performed and analyzed. The complexes were solvated in dodecahedron water boxes using a four-point TIP4P rigid water model with at least 1 nm of solvation on all sides, and neutralization was achieved by adding Na^+^ ions. The particle mesh Ewald (PME) summation method was used for the treatment of long-range interactions with all bonds constrained using the LINCS algorithm. Further, the energy minimization of the system was carried out using a steepest descent method at a temperature of 310 K and one atmospheric bar pressure via a V-rescale thermostat and Parrinello–Rahman barostat implementation. The conformations were obtained at intervals of 10 ps throughout the 25 ns trajectory. Post-simulation, energy minimization and trajectory analysis showed that the complex initially showed 2 Å deviations but achieved stability later for the top three selected complexes (Figure 2). Residue-based root mean square fluctuation (RMSF) analysis of antigen–antibody docked complexes was performed to understand the flexibility of each residue, as depicted in Figure 2a. RMSF values for all docked complexes showed large fluctuations (0.33–1.67 nm) for the initial 20 residues in each case due to the unavailability of structural information for the target proteins (Figure 2d). Further, lesser fluctuations at the binding and active sites indicated the intactness and rigidity of the binding cavities. gmx_gyrate was used to calculate Rg values indicating the compactness and structural changes of the docked complexes. Rg is a measure of the mass of atoms with respect to the center of mass of complexes (Figure 2b). Average Rg values for Bla g 1.0101, Per a 1.0102, and Per a 7 ranged between 2.36 and 2.66 nm, 2.29 and 2.89 nm, and 2.44 and 2.56 nm, respectively, with no fluctuations after 25,000 ps. Further, Rg values correlated with RMSD values for backbone Cα atoms, validating the stability of the prioritized antigen–antibody complexes. These results indicate the suitability of the prioritized allergenic protein sequences among all the proteomes for further investigation in bench-top experiments.

## 4. Conclusions

*Blattella germanica* allergens appear to be about equally concentrated in homes in Saudi Arabia, even though the prevalence of the Bla g 2 allergen was found to be slightly greater in patients’ homes. As there is little information available regarding cockroach-related allergens, it is essential to investigate this issue in detail to empower ourselves with a remedy in advance. The current study intended to find immunodominant peptides that may be employed in the future to develop a universal peptide vaccine to treat cockroach-related illnesses. This will aid in eradicating future possibilities of asthma and allergenic rhinitis. In this study, an immunoinformatics approach was applied to evaluate the immunogenicity of prioritized proteins. Research incorporating experimental confirmation of these predicted epitopes is necessary to ensure the capabilities of B-cell and T-cell stimulations for their efficient use as vaccine candidates and as diagnostic agents against cockroaches.

## Figures and Tables

**Figure 1 molecules-27-08740-f001:**
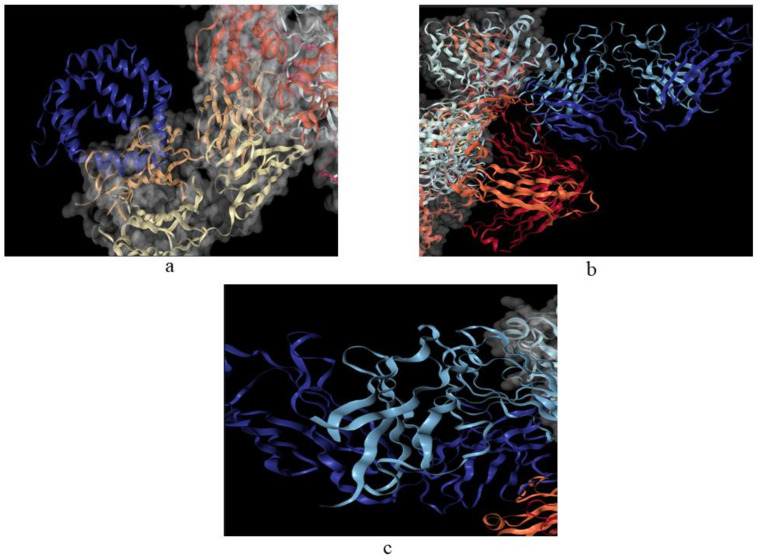
Visualization of docked complexes of (**a**) Blag 1.0101-IgE (−19.08 kcal/mol), (**b**) Per a 1.0102-IgE (−21.33 kcal/mol), and (**c**) Per a 7-IgE (−19.71 kcal/mol). IgE is represented as a complex molecule, and selected protein allergens are visualized as blue polypeptide chains.

**Figure 2 molecules-27-08740-f002:**
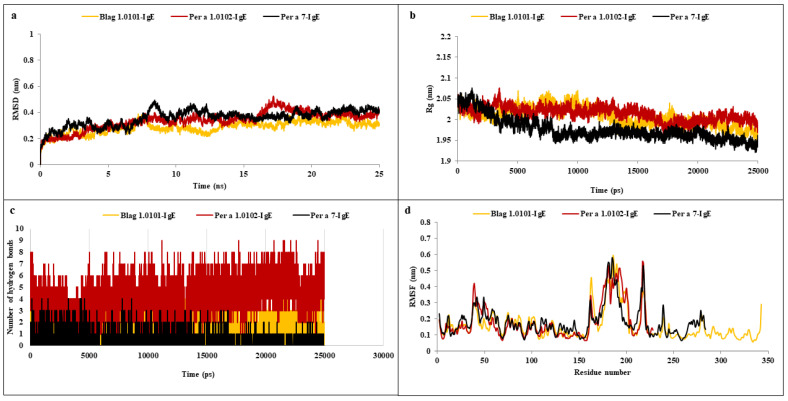
(**a**) RMSD, (**b**) Rg, (**c**) hydrogen bond, and (**d**) RMSF analyses of Blag 1.0101-IgE, Per a 1.0102-IgE, and Per a 7-IgE.

**Table 1 molecules-27-08740-t001:** Physiochemical characterization of the protein sequences from date palms and cockroaches.

S. No.	Protein Seq.	Uniprot /NCBI	No. of Amino Acids	Molecular Wt.	pI	Ext. Cof.	Instability Index	Aliphatic Index	GRAVY
		***Blattella germanica* Allergens**							
1	Bla g 1.0101	Q9UAM5	412	45817.74	4.49	14900	38.52	129.54	0.176
2	Bla g 1.0201	O96522	492	55507.76	4.70	19370	46.59	120.51	0.046
3	Bla g 2	P54958	352	38557.88	5.28	35005	26.72	91.85	0.076
4	Bla g 3	D0VNY7	657	78737.11	6.44	123110	44.65	65.21	−0.665
5	Bla g 4	P54962	182	20927.45	6.48	27640	28.02	72.42	−0.567
6	Bla g 5	O18598	204	23333.71	6.32	45380	31.54	80.39	−0.490
7	Bla g 6.0101	Q1A7B3	151	17216.22	3.96	2980	35.44	86.56	−0.388
8	Bla g 6.0201	ABB89297	151	17095.14	3.96	3105	31.48	87.22	−0.269
9	Bla g 6.0301	ABB89298	154	17749.86	4.10	2980	53.85	81.10	−0.496
10	Bla g 7	Q9NG56	284	32837.84	4.72	6085	44.60	81.94	−0.973
11	Bla g 8	A0ERA8	195	21180.40	4.44	6990	38.88	67.28	−0.567
		***Periplaneta americana* Allergens**							
1	Per a 1.0101	Q9TZR6	231	26222.88	4.46	10430	57.52	114.46	−0.014
2	Per a 1.0102	O18535	228	25791.39	4.44	10430	56.32	115.96	0.004
3	Per a 1.0103	O18530	395	44610.99	4.55	24870	52.06	112.10	0.017
4	Per a 1.0104	O18528	274	31142.54	4.45	18910	55.73	114.64	−0.020
5	Per a 1.0201	O18527	446	50547.54	5.31	16390	56.57	108.05	−0.179
6	Per a 3.0101	Q25641	685	81175.35	6.25	132385	41.56	67.11	−0.612
7	Per a 3.0201	Q94643	631	75511.84	6.61	120590	44.03	64.07	−0.737
8	Per a 3.0202	Q25640	470	561888.05	7.02	85625	42.03	64.26	−0.782
9	Per a 3.0203	Q25639	393	46746.52	6.52	65225	46.06	66.92	−0.749
10	Per a 6	Q1M0Y3	151	17130.99	3.84	2980	24.97	87.22	−0.377
11	Per a 7	Q9UB83	284	32776.81	4.69	6085	45.04	81.58	−0.920
12	Per a 7.0102	P0DSM7	284	32793.87	4.72	4595	45.01	83.31	−0.924
13	Per a 9	B9VAT1	356	39735.09	5.58	36120	30.93	80.56	−0.413
14	Per a 10	Q1M0X9	256	26652.14	4.89	34420	25.77	84.49	0.273
		***Phoenix dactylifera* Allergens**							
1	XP_008803750.1	P33050	176	18902.39	5.93	11845	34.39	73.75	−0.109
2	XP_008780644.1	A0A0S3B0K0	182	20044.54	6.25	18910	27.85	74.01	−0.352
3	XP_008782456.1	-	201	22566.84	6.12	23295	51.48	70.75	−0.286
4	AGE46030.1	-	361	41027.22	5.66	68465	33.46	81.05	−0.247
5	XP_008796227.1	-	529	59084.12	5.93	94350	35.72	77.75	−0.364
6	XP_010911620.1	O81355	309	33749.62	6.00	17420	30.12	98.45	−0.013
7	YP_005090378.1	-	509	55292.25	6.02	25580	36.20	95.07	−0.129
8	XP_008781205.1	-	396	43329.14	5.71	30410	20.56	81.46	−0.315
9	XP_008811417.1	Q2UMD5	823	92522.51	5.89	188840	31.89	71.56	−0.448

**Table 2 molecules-27-08740-t002:** Listing of primary protein sequences in functional modules.

S. No.	Protein Allergen	Uniprot /NCBI	Functional Class	Functionality
1	Bla g 1.0101	Q9UAM5	Metabolism molecule	-
2	Bla g 1.0201	O96522	Virulence factors	-
3	Bla g 2	P54958	Cellular process	Acid proteases (SSF50630)
4	Bla g 3	D0VNY7	Cellular process	E set domains (SSF81296); Hemocyanin, N-terminal domain (SSF48050); Di-copper centre-containing domain (SSF48056)
5	Bla g 4	P54962	Cellular process	Lipocalins (SSF50814)
6	Bla g 5	O18598	Metabolism molecule	Thioredoxin-like (SSF52833); GST C-terminal domain-like (SSF47616)
7	Bla g 6.0101	Q1A7B3	Information and storage	EF-hand (SSF47473)
8	Bla g 6.0201	ABB89297	Cellular process	-
9	Bla g 6.0301	ABB89298	Information and storage	-
10	Bla g 7	Q9NG56	Information and storage	-
11	Bla g 8	A0ERA8	Cellular process	EF-hand (SSF47473)
12	Per a 1.0101	Q9TZR6	Metabolism molecule	-
13	Per a 1.0102	O18535	Metabolism molecule	-
14	Per a 1.0103	O18530	Metabolism molecule	-
15	Per a 1.0104	O18528	Metabolism molecule	-
16	Per a 1.0201	O18527	Metabolism molecule	-
17	Per a 3.0101	Q25641	Metabolism molecule	E set domains (SSF81296); Hemocyanin, N-terminal domain (SSF48050); Di-copper centre-containing domain (SSF48056)
18	Per a 3.0201	Q94643	Information and storage	E set domains (SSF81296); Hemocyanin, N-terminal domain (SSF48050); Di-copper centre-containing domain (SS8F480596)
19	Per a 3.0202	Q25640	Cellular process	E set domains (SSF81296); Di-copper centre-containing domain (SSF48056)
20	Per a 3.0203	Q25639	Cellular process	E set domains (SSF81296); Di-copper centre-containing domain (SSF48056)
21	Per a 6	Q1M0Y3	Cellular process	EF-hand (SSF47473)
22	Per a 7.0101	Q9UB83	Information and storage	-
23	Per a 7.0102	P0DSM7	Information and storage	-
24	Per a 9	B9VAT1	Cellular process	Guanido kinase N-terminal domain (SSF48034); Glutamine synthetase/guanido kinase (SSF55931)
25	Per a 10	Q1M0X9	Cellular process	Trypsin-like serine proteases (SSF50494)
26	XP_008803750.1	P33050	Metabolism molecule	Enolase C-terminal domain-like (SSF51604); Enolase N-terminal domain-like (SSF54826)
27	XP_008780644.1	A0A0S3B0K0	Metabolism molecule	RmlC-like cupins (SSF51182)
28	XP_008782456.1	-	Cellular process	-
29	AGE46030.1	-	Metabolism molecule	-
30	XP_008796227.1	-	Virulence factors	-
31	XP_010911620.1	O81355	Cellular process	NAD(P)-binding Rossmann-fold domains (SSF51735)
32	YP_005090378.1	-	Cellular process	-
33	XP_008781205.1	-	Cellular process	-
34	XP_008811417.1	Q2UMD5	Cellular process	(Trans)glycosidases (SSF51445); Beta-galactosidase LacA, domain 3 (SSF117100); Galactose-binding domain-like (SSF49785)

**Table 3 molecules-27-08740-t003:** AlgPred: Prediction of allergenic proteins and mapping of IgE epitopes.

S. No.	Protein Seq.	IgE Epitope	Sequence Matched	Pos.	PIDs	ARPs
1	Bla g 1.0101	* LIRALFGL * LIRALFGL * LIRALFGL	*LIRALFGL *LIRALFGL *LIRALFGL	19 211 403	100 100 100	VDHFIQLIRALFGLS---RAARNLQDD
2	Bla g 1.0201	The protein sequence does not contain an experimentally proven IgE epitope				IHSIIGLPPFVPPSRRHARRGVGI
3	Bla g 2	The protein sequence does not contain an experimentally proven IgE epitope				IEDSLTISNLTTSQQDIVLADELS
4	Bla g 3	The protein sequence does not contain an experimentally proven IgE epitope				LYTYFEHFEHSLGNAMYIGKLEDL
5	Bla g 4	The protein sequence does not contain an experimentally proven IgE epitope				DALVSKYTDSQGKNRTTIRGRTKF
6	Bla g 5	The protein sequence does not contain an experimentally proven IgE epitope				YHYDADENSKQKKWDPLKKETIPY
7	Bla g 6.0101	The protein sequence does not contain an experimentally proven IgE epitope				Not found
8	Bla g 6.0201	The protein sequence does not contain an experimentally proven IgE epitope				Not found
9	Bla g 6.0301	The protein sequence does not contain an experimentally proven IgE epitope				Not found
10	Bla g 7	* AQLLAEEADRKYD * EKYKSITDELDQTFS * ELVNEKEKYKSITDE * ESKIVELEEELRVVG * MQQLENDLDQVQESLLK * QKLQKEVDRLEDELV * RIQLLEEDLERSEER * RSLSDEERMDALENQ * VAALNRRIQLLEEDL * VDRLEDELVNEKEKY	* ARFMAEEADKKYD * EKYKYICDDLDMTFT * ELVHEKEKYKYICDD * ESKIVELEEELRVVG * IQQIENDLDQTMEQLMQ * QKLQKEVDRLEDELV * RIQLLEEDLERSEER * KGLADEERMDALENQ * VAALNRRIQLLEEDL * VDRLEDELVHEKEKY	151 265 259 187 50 247 91 133 85 253	69.23 66.66 73.33 100 58.82 100 100 80 100 93.33	GESKIVELEEELRVVGNNLKSLEV
11	Bla g 8	The protein sequence does not contain an experimentally proven IgE epitope				Not found
12	Per a 1.0101	* LIRALFGL * LIRALFGL	* LIRALFGL * LIRSLFGL	35 223	100 87.5	FKNFLNFLQTNGLNAIEFLNNIH
13	Per a 1.0102	* LIRALFGL * LIRALFGL	* LIRALFGL * LIRSLFGL	32 220	100 87.5	FKNFLNFLQTNGLNAIEFLNNIH
14	Per a 1.0103	* LIRALFGL * LIRALFGL	* LIRALFGL * LIRSLFGL	199 387	100 87.5	AYLHADDFHKIITTIEA
15	Per a 1.0104	* LIRALFGL	* LIRALFGL	78	100	LPEDLQDFLALIPIDQILAIAAD
16	Per a 1.0201	* LIRALFGL	* LIRALFGL	101	100	FKNFLNFLQTNGLNAIEFLNNIH
17	Per a 3.0101	The protein sequence does not contain an experimentally proven IgE epitope				SVFHFYRLLVGHVVDPYHKNGLAP
18	Per a 3.0201	The protein sequence does not contain an experimentally proven IgE epitope				RLNHKPFTYNIEV
19	Per a 3.0202	The protein sequence does not contain an experimentally proven IgE epitope				RLNHKPFTYNIEV
20	Per a 3.0203	The protein sequence does not contain an experimentally proven IgE epitope				RLNHKPFTYNIEV
21	Per a 6	The protein sequence does not contain an experimentally proven IgE epitope				Not found
22	Per a 7	* AQLLAEEADRKYD * EKYKSITDELDQTFS * ELVNEKEKYKSITDE * ESKIVELEEELRVVG * MQQLENDLDQVQESLLK * QKLQKEVDRLEDELV * RIQLLEEDLERSEER * RSLSDEERMDALENQ * VAALNRRIQLLEEDL * VDRLEDELVNEKEKY	* ARFMAEEADKKYD * EKYKYICDDLDMTFT * ELVHEKEKYKYICDD * ESKIVELEEELRVVG * IQQIENDLDQTMEQLMQ * QKLQKEVDRLEDELV * RIQLLEEDLERSEER * KGLADEERMDALENQ * VAALNRRIQLLEEDL * VDRLEDELVHEKEKY	151 265 259 187 50 247 91 133 85 253	69.23 66.66 73.33 100 58.82 100 100 80 100 93.33	GESKIVELEEELRVVGNNLKSLEV
24	Per a 7.0102	* AQLLAEEADRKYD * EKYKSITDELDQTFS * ELVNEKEKYKSITDE * ESKIVELEEELRVVG * MQQLENDLDQVQESLLK * QKLQKEVDRLEDELV * RIQLLEEDLERSEER * RSLSDEERMDALENQ * VAALNRRIQLLEEDL * VDRLEDELVNEKEKY	* ARFMAEEADKKYD * EKYKYICDDLDMTFT * ELVHEKEKYKYICDD * ESKIVELEEELRVVG * IQQIENDLDQTMEQLMQ * QKLQKEVDRLEDELV * RIQLLEEDLERSEER * KGLADEERMDALENQ * VAALNRRIQLLEEDL * VDRLEDELVHEKEKY	151 265 259 187 50 247 91 133 85 253	69.23 66.66 73.33 100 58.82 100 100 80 100 93.33	GESKIVELEEELRVVGNNLKSLEV
23	Per a 9	The protein sequence does not contain an experimentally proven IgE epitope				KLPKLAANREKLEEVAAKFSLQVR
24	Per a 10	The protein sequence does not contain an experimentally proven IgE epitope				CNGDSGGPLVSANRKLTGIVSWG
25	XP_008803750.1	The protein sequence does not contain an experimentally proven IgE epitope				IQGQVYCDTCRAGFITELSEFI
26	XP_008780644.1	The protein sequence does not contain an experimentally proven IgE epitope				Not found
27	XP_008782456.1	The protein sequence does not contain an experimentally proven IgE epitope				Not found
28	AGE46030.1	The protein sequence does not contain an experimentally proven IgE epitope				Not found
29	XP_008796227.1	The protein sequence does not contain an experimentally proven IgE epitope				Not found
30	XP_010911620.1	The protein sequence does not contain an experimentally proven IgE epitope				MVSIFHTIYVKGDQTNFQIGP
31	YP_005090378.1	The protein sequence does not contain an experimentally proven IgE epitope				Not found
32	XP_008781205.1	The protein sequence does not contain an experimentally proven IgE epitope				Not found
33	XP_008811417.1	The protein sequence does not contain an experimentally proven IgE epitope				Not found
34	XP_008803750.1	The protein sequence does not contain an experimentally proven IgE epitope				Not found

** PID: Percent of identity, ARPs: allergen representative proteins, Pos: Protein composition.*

**Table 4 molecules-27-08740-t004:** Protein allergenicity potential predictions (AllerCatPro 2.0 Results).

S. No.	Protein Allergen	IgE Prevalence			Similarity to Allergen and Resulting Predicted Evidence for Allergenicity	
		Uniprot /NCBI	Pfam	InterPro		
1	Bla g 1.0101	Q9UAM5	PF06757	IPR010629	1052	Strong evidence
2	Bla g 7	Q9NG56	PF00261	IPR000533	-	Strong evidence
3	Per a 1.0101	Q9TZR6	PF06757	IPR010629	15	Strong evidence
4	Per a 1.0102	O18535	PF06757	IPR010629	211	Strong evidence
5	Per a 1.0103	O18530	PF06757	IPR010629	211	Strong evidence
6	Per a 1.0104	O18528	PF06757	IPR010629	15	Strong evidence
7	Per a 1.0201	O18527	PF06757	IPR010629	211	Strong evidence
8	Per a 7	Q9UB83	PF00261	IPR000533	17010	Strong evidence
9	Per a 7.0102	P0DSM7	-	-	-	Strong evidence

**Table 5 molecules-27-08740-t005:** Secondary structure analysis of nine screened protein sequences.

S. No.	Protein Seq.	Alpha Helix	Extended Strand	Beta Turn	Random Coil
1	Bla g 1.0101	305 is 74.03%	2 is 0.49%	17 is 4.13%	88 is 21.36%
2	Bla g 7	280 is 98.59%	0 is 0.00%	1 is 0.35%	3 is 1.06%
3	Per a 1.0101	181 is 78.35%	0 is 0.00%	7 is 3.03%	43 is 18.61%
4	Per a 1.0102	176 is 77.19%	0 is 0.00%	8 is 3.51%	44 is 19.30%
5	Per a 1.0103	279 is 70.63%	9 is 2.28%	17 is 4.30%	90 is 22.78%
6	Per a 1.0104	208 is 75.91%	4 is 1.46%	9 is 3.28%	53 is 19.34%
7	Per a 1.0201	319 is 71.52%	6 is 1.35%	18 is 4.04%	103 is 23.09%
8	Per a 7	278 is 97.89%	0 is 0.00%	1 is 0.35%	5 is 1.76%
9	Per a 7.0102	281 is 98.94%	0 is 0.00%	0 is 0.00%	3 is 1.06%

## Data Availability

The authors confirm that the data supporting the findings of this study are available within the article, including the Appendix A. Raw data that support the findings of this study are available from the corresponding author upon request.

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
