# Peer review of "In Silico Comparative Exploration of Allergens of Periplaneta americana, Blattella germanica and Phoenix dactylifera for the Diagnosis of Patients Suffering from IgE-Mediated Allergic Respiratory Diseases"

_molecules, 2022, doi:10.3390/molecules27248740_

Round 1

Reviewer 1 Report (Previous Reviewer 2)

1.     Since the allergen protein was retraveled from the Uniprot, the structural information have been stated, then why used the MODELLER software?

2.     What kind of IgE was used for the docking studies? The information should be added

3.     The Uniprot ID for the allergen should be added in the Table 1, and the structure accuracy

4.     For the table 2, the functions of the protein in the Uniprot also be marked, then the information should also be added

5.     The results of the secondary structure prediction should be added

6.     The 3D structures of the proteins and the validation results should be added, and why not choose the Per a allergens

7.     No useful discussion in the result section

8.     What is the potential IgE epitope

Author Response

Reviewer 2 Report (New Reviewer)

Kausar et al. conducted an in silico study on IgE-mediated allergens by using various web servers and molecular simulation method. Several peptides were identified to have potential binding affinity with IgE. Finding immunodominant peptides is important to future development of vaccine to treat cockroaches-related illness, aiding in eradicating the future possibility of asthma and allergenicrhinitis. I have the following concerns:

Major concerns:

1. The results part is very weak, and could not give sufficient supports to conclusions. The authors should carried out more analysis to make the outcomes solid. For instance, they can do binding free energy calculation on the 3 selected IgE complexes, analyzing the key residues via energy decomposition. Also they can do SASA calculation to examine the hydrophobicity. Meantime, due to the flexibility of peptides, the initial coordinates affects the results greatly. So the selection of antigen antibody complexes should be careful and need more verification. Maybe MD simulations should be performed on more complexes to obtain potential candidates.

2. MD simulations of the method parts: The description of MD simulation is confusing. It is said "docked complexes were subjected to 309K for 25ns", and "Non-hydrogen solute atoms were restricted to 300 K and 1bar for 40ns to attain equilibrium state". Then what on earth is the temperature and timescale for the production MD, i.e., on which trajectory did the authors carry out the analysis?

3. P8. Antigen antibody docking: It can be seen that the 3D structure of selected 3 sequences were built using homology modeling method based on three template proteins with PDB code 4JRB, 7KO4 and 6X5Z. But it is not stated in the text how the IgE structure was obtained? from PCSB PDB or homology modeling?

4. P9. Figure 1: The resolution of the figure is poor, and can't provide efficient information. The mentioned residues that involved in hydrogen bonds should be shown. The residues of peptide and IgE can be depicted using different color schemes.

5. P10. Figure 2(b): The simulation is 25 ns long, then how can the author draw the conclusion "with no fluctuations after 25000ps"? Figure 2(d): I can't see " the RMSF values for all docked complexes had large fluctuation (0.33-1.67nm) for the initial 20 residues in each case due to ...". The largest fluctuation is about 0.6 nm or so in this figure, and the largest fluctuation occurs for central residues (180~220). Meantime, the correlation between hydrogen bond number and the docking energies can be stated in text.

Minor concerns:

1. Abstract: Only why and how to do the study were contained, and meaningful results should be included.

2. P9. MD simulation: The initial description of MD procedure is confusing as in the method part. For example, energy minimization was carried out at 310K this time. Only NVT/NPT simulations employ thermostat/barostat.

3. P10. The legend in the figure "Per a 1.01.3-IgE" should be " Per a 1.01.2-IgE ".

4. P4. 2. Physiochemical parameter evaluation of the above allergens: "..., molecular and GRAVY..." should be "..., molecular weight and GRAVY...".

5. P8. 6. Secondary structure prediction: "This infers that hydrogens are mainly responsible..." should be "This infers that hydrogen bonds are mainly responsible...".

Round 2

Reviewer 1 Report (Previous Reviewer 2)

1 the selected proteins should be listed in the 1 section of result, and whether all the proteins in the UniPort have been selected in the Table 1, by the way, what is the principle for the protein selection?

2 for the docking section, little information were discussed about the IgE epitopes

3 the abstract should be revised for the novelty of the paper

Author Response

Reviewer 2 Report (New Reviewer)

see the attachment.

Author Response

This manuscript is a resubmission of an earlier submission. The following is a list of the peer review reports and author responses from that submission.

Round 1

Reviewer 1 Report

This paper reports an attempt to identify the allergens for allergic respiratory diseases using docking simulation, molecular dynamics (MD) simulation, and several public databases. Such an attempt is very challenging and important in the research of allergies. However, this manuscript does not satisfy the format of a scientific paper and is incomplete, so the authors must revise the manuscript substantially before the scientific review. Specific comments are listed below.

Comment 1:
Figure 2 is not given while it is referred to in the main text.

Comment 2:
Figure 3 is not given while it is referred to in the main text.

Comment 3:
A conclusion or summary section should be given at the end.

Comment 4:
The authors state “Therefore, in this study, only stable amino acids having an instability index <40 is selected for further allergenicity and functional prediction.” in lines 172 to 174. But even after that, all 34 candidates, which includes the molecules whose instability index is larger than 40, are still examined. 

Comment 5:
In Table 2, two “Per a 1.0103” are listed in the column of Metabolism molecule.

Comment 6:
There is a description “This machine learning based tool identifies the active participation of the primary protein sequences into metabolism molecule (33.33%), virulence factors (6.01%), cellular processes (40.45%) and information and storage (20%)” in lines 180 to 181. How were these percentages calculated? In my understanding, they should be as followed:
Metabolism molecule: 11/34 = 0.323529 → 32.35%
Virulence factors: 2/34 = 0.058834 → 5.88%
Cellular processes: 15/34 = 0.441176 → 44.12%
Information and storage: 6/34 = 0.176470 → 17.65%
Why are the values slightly different?

Comment 7
I think Figure 1 is not needed because the percentages are given in the main text.

Comment 8:
I think the column of “Functional Class” in Table 2 is not needed.

Comment 9:
Serial numbers of the proteins are different in Tables 1, 3, and 4. I think it is better to be unified.

Comment 10:
In Tables 1 and 3, the notation of “Per a 7” is used, but in Tables 2 and 4, the notation of “Per a 7.0101” is used. The same notation should be used throughout the paper.

Comment 11:
More detailed information on the homology modeling should be given. What proteins were used as the templates? What was the homology of the sequences?

Comment 12:
For the docking simulation, only the binding energies of the selected three complexes are given. But the binding energies of the other six complexes should also be given. The difference in the binding energies between the selected and not selected complexes are important information for the readers.

Comment 13:
It is difficult to accurately review the MD simulation because the data of the MD simulation (Figure 3) is not given at this stage. But generally, 25ns trajectory seems short for MD simulation f a protein-protein complex.

Comment 14:
The title of the manuscript says "IgE Mediated", but all the complex screened from the docking calculation were complex with IgM. Is that no problem?

Reviewer 2 Report

The experimental design and experimental content generally had no innovation because there has been a lot of research on the related content. The introduction part is not comprehensive enough. In the Result and Discussion section, the author shows us a lot of results, however, there isn't an adequate discussion of the results here.

Major comment

Comment 1: In the Abstract section, more result section should be added.

Comment 2: In the Introduction section, line 67 to 73, why introduced Per a 10?

Comment 3: Why are the Allergens of Periplaneta americana, Blattella germanica, and Phoenix dactylifera selected for comparative in silico? In the Introduction section, the author did not well highlight the innovation and significance of the in-silico analysis

Comment 4: In the study, in-silico analysis was applied for allergens of Periplaneta

americana, Blattella germanica and Phoenix dactylifera. However, in the Introduction section, the author did not introduce Phoenix dactylifera.

Comment 5: Why the subcellular localization is analyzed by different tools for different allergens?

Comment 6: Flexible molecular docking should be considered in Antigen-Antibody docking studies.

Comment 7: The first part of the result should be removed in the Result and Discussion.

Comment 8: In the Result and Discussion section, there isn't an adequate discussion of the results here.

Comment 9: It may be better if the article with a conclusion section to summarize the work and the meaning of the work.  

Comment 10: The reference format is not uniform, please check.

Minor comment:

Comment 1: Line 33, please note the canonical use of punctuation.

Comment 2: Lines 168, “GRAVY”? Please full the name.

Comment 3: In Table 3, what does “Pos., PIDs, ARPs” mean?

Comment 4: Line 245, there should be a space between “1.67” and “nm”, please check the full text.